# Chiral Recognition of Homochiral Poly (amidoamine) Dendrimers Substituted with *R*- and *S*-Glycidol by Keratinocyte (HaCaT) and Squamous Carcinoma (SCC-15) Cells In Vitro

**DOI:** 10.3390/polym13071049

**Published:** 2021-03-27

**Authors:** Małgorzata Malinga-Drozd, Łukasz Uram, Konrad Wróbel, Stanisław Wołowiec

**Affiliations:** 1Department of Biochemistry and General Chemistry, Medical College, Rzeszów of University, 1a Warzywna Str., 35-310 Rzeszów, Poland; mmalinga@wp.pl (M.M.-D.); konradwrobel300@gmail.com (K.W.); 2Department of Inorganic and Analytical Chemistry, Faculty of Chemistry, Rzeszów University of Technology, 6 Powstańców Warszawy Ave., 35-959 Rzeszów, Poland; luram@prz.edu.pl

**Keywords:** homochiral dendrimer, chiral biorecognition, confocal microscopy, glycidol, polyamidoamine dendrimer, toxicity

## Abstract

The generation 2 and 3 poly(amidoamine) dendrimers (PAMAM G2 and G3) were converted into N-(2,3-dihydroxy)propyl derivatives by the addition of enantiomerically pure *S*- and *R*-glycidol. The homochiral dendrimers bind to HaCaT and SCC-15 cell membranes with an *R*/*S* glycidol enantioselectivity ratio of 1.5:1, as was quantitatively determined by fluorescence microscopy and visualized by confocal microscopy. Fully substituted G2 and G3 dendrimers were equipped with 32 and 64 N-(2,3-dihydroxy)propyl residues and showed effectively radial symmetry for homochiral derivatives in ^13^C NMR spectrum in contrary to analogs obtained by reaction with *rac*-glycidol. The sub-stoichiometric derivatives of G2 and G3 were also obtained in order to characterize them spectroscopically. The homochiral dendrimers were labeled with two different fluorescent labels, fluorescein, and rhodamine B, using their isothiocyanates to react with G2 and G3 followed by the addition of *S*- and *R*-glycidol. Obtained fluorescent derivatives were deficiently filled with N-(2,3-dihydroxy)propyl substituents due to steric hindrance imposed by the attached label. Nevertheless, these derivatives were used to determine their ability to bind to the cell membrane of human keratinocytes (HaCaT) and squamous carcinoma cells (SCC-15). Confocal microscopy images obtained from cells treated with variously labeled conjugates and fluorescence analysis with fluorescence reader allowed us to conclude that *R*-glycidol derivatives were bound and entered the cells preferentially, with higher accumulation in cancer cells. The G3 polyamidoamine (PAMAM)-based dendrimers were taken up more efficiently than G2 derivatives. Moreover, *S*- and *R*-glycidol furnished dendrimers were highly biocompatible with no toxicity up to 300 µM concentrations, in contrast to the amine-terminated PAMAM analogs.

## 1. Introduction

Polyamidoamine (PAMAM) dendrimers were synthesized in 1985 by Tomalia et al. [1]. Since that time, PAMAM dendrimers became often explored macromolecular reagents due to their strictly defined molecular weight, radial symmetry, and availability of terminal functional groups on the surface [2]. Full generation PAMAM dendrimers provide amine groups that can be functionalized with various substituents, including drug molecules. Therefore, they are continuously tested, especially dendrimers of generation 3, 4, and 5 (G3, G4, G5), as drug carriers and gene delivery systems [3,4]. The toxicity of PAMAM G2-G6 dendrimers is rather low, while 0.5 μM concentration near cell surface already enables to observe cell internalization of these molecules [5]. Therefore, PAMAM G3, G4, and G5 dendrimers with 32, 64, and 128 amine groups, respectively, were used to covalently bind anticancer drug molecules such as methotrexate [6], daunorubicin [7], or paclitaxel [8] to obtain highly drug-loaded conjugates. Additionally, the hydrophilic properties of these macromolecular drug carriers can be tuned by terminal amine group acylation or polyhydroxylation with glycidol [7,8,9] or carbohydrate lactones [6,10,11]. Thus, PAMAM chiral dendrimers can be obtained that are surface-modified with various substituents, derived from D-gluconolactone [10], D-glucoheptono-1,4-lactone [6,12], amino acids [13], and by polyethylene glycol (PEG)ylation [14,15]. Generally, the chirality in dendritic architecture is achievable using chiral core and build-up of dendritic arms (class 1), chiral building blocks (class 2), and encountered above end-group chiral substituents (class 3) [16].

The example of class 1 dendrimers are BINAP-cored and polyphenylether-expanded dendrimers, which were shown to provide catalytic space for hydrogenation of various substrates, i.e., alkenes, such as 2-(4-isobutylphenyl)acrylic acid or acetamidocinnamic acid, and acetophenones, which were reduced into corresponding chiral products with high chemoselectivity and >90% enantiomeric excess (ee) in presence of catalytically active Ru(I) and Rh(II) organometallic complexes [17].

The example of class 2 chiral dendrimers constructed of chiral vicinal diol subunits linked and branched by phenyl groups are polyhydroxylated dendrimers obtained in 2000 by McElhanon and McGrath [18] and PAMAM type dendrimers constructed with optically pure (-)1,2-diaminopropane instead of ethylenediamine [19]. Benzyl group-protected PAMAM dendrimers of generation 2 were used further to encapsulate Pd and Rh nanoparticles (NPs). The encapsulates of Pd and Rh in PAMAM-32Bn had 1.7 nm diameter and showed a positive Cotton effect in circular dichroism spectrum with maximum red shifted by ca 25 and 50 nm in comparison with Cotton effect of the host (ca. 240 nm). This behavior evidenced the presence of PdNP and RhNP within the voids of dendrimer, not on the surface. This opened a door for Pd(0) catalysis inside chiral dendritic voids for such processes such as C-C cross coupling reactions, which were successfully performed within non-chiral 3.5 nm sized Pd@PAMAM G3 encapsulates [20].

Chiral PAMAM G3 and G4 dendrimers of class 3, modified by amidation of primary amine groups with D-gluconolactone were demonstrated to induce the asymmetric reduction of prochiral ketones into chiral alcohols with NaBH_4_ with high yield (>90%) and enantiomeric excess (ee), depending on dendrimer generation and solvent (THF or H_2_O) [10].

Another example of chiral PAMAM (G1-G3) dendrimer (class 3) was obtained as multicenter Ti (IV) dendrimer by covalent attachment of (-N-(3,5-di-tert-butylsalicylidene)-N’-[3-tert-butyl-5-chloromethylsalicylidene]-1,2-cyclohexanediamine, chiral salen type ligand via imidazolyl linkers. Dimeric Ti(IV)-(µ-O)_2_-Ti(IV) centers cooperatively oxidized methyl R-phenyl sulfide into sulfoxide with 90% chemoselectivity and > 80% ee, where R was variable substituent on phenyl group [21].

Chiral space inside the dendrimer voids is currently exploited for catalytic purposes. As exemplified above, the chiral substituents on the periphery of dendrimers can operate as asymmetry inductors and be recognized by biological membrane receptors, especially if chiral inductors are enantiomerically pure amino acids or specific carbohydrates. The simplest chiral substituent, readily introduced into peripheral amine groups of PAMAM dendrimers is 2, 3-dihydroxypropyl residue, as it was demonstrated earlier [6]. PAMAM dendrimers react with glycidol at ambient temperature. Primary amine groups are thus converted into bis-2,3-dihydroxyalkylamine groups. The products of substitution are easy to purify and can be fully characterized by NMR spectroscopy and other methods. This has been carried out in a complete and elegant way by Shi et al. in 2005 for PAMAM G1 dendrimer [22]. In the study, the racemic glycidol was used and PAMAM G1 substituted with an average of 14 equivalents of glycidol was characterized as an incomplete derivative. Additionally, the C-13 NMR spectra showed a complicated pattern for 2, 3-dihydroxyalkyl resonances, i.e., more than two sets of resonances, suggesting that symmetry-dependent species recognize not only geminal 2-*R*- or 2-*S*, 3-dihydroxypropyl substituents, but the next arm of the dendrimer is involved in symmetry recognition.

In our way to construct drug delivery systems (DDS) based on dendrimers, we used PAMAM G3 dendrimer totally covered with racemic glycidol as a central dendrimer to which low generation G0 PAMAM dendrimers were attached covalently. The *rac*-glycidol-covered core G3 was able to bind up to 12 G0, and this megamer was demonstrated as an efficient DDS for nimesulide, the non-steroidal anti-inflammatory drug [23]. In order to improve this DDS, we prepared the PAMAM G3 and G2 totally and partially substituted with optically pure *R*- and *S*-glycidol, studied them by NMR spectroscopy, and finally tested their ability to bind to the cell membrane and enter the cell on two various cell lines—normal keratinocytes (HaCaT) and squamous carcinoma cells (SCC-15). The cell cultures were monitored by fluorescence microscopy and confocal microscopy with two fluorescent labels—fluorescein and rhodamine B isothiocyanates. Surprisingly, we observed some enantiodiscrimination of homochiral dendrimers in cell membrane interaction by both types of cells and generation-dependent internalization.

## 2. Experimental

### 2.1. Reagents

Ethylenediamine, methyl acrylate, racemic glycidol (96%), *R*- and *S*-glycidol (both 98% purity), and fluorescein isothiocyanate (isomer I, 90%), and rhodamine B isothiocyanate were purchased from Merck KGaA (Darmstadt, Germany). PAMAM dendrimers were synthesized by alternate addition of methyl acrylate into amine groups, starting from ethylenediamine core, followed by condensation with ethylenediamine, purified intermediates at every step according to the protocol of Tomalia [1], and stored as 20 weight% solutions in methanol.

Human squamous carcinoma cells (SCC-15), penicillin, and streptomycin solutions were obtained from American Type Culture Collection (ATCC, Manassas, VA, USA). Human immortalized keratinocytes (HaCaT) were purchased from Cell Lines Service (Eppelheim, Germany). Dulbecco’s modified Eagle’s medium (DMEM), Dulbecco’s, modified Eagle’s medium F12 (DMEM-F12), and fetal bovine serum (FBS) were obtained from Corning (NewYork NY, USA). Trypsin–EDTA solution, phosphate-buffered saline (PBS) with and without magnesium and calcium ions, neutral red, 0.4% trypan blue solution, sterile syringe filters, 0.22 µm were provided by Sigma–Aldrich (St Louis, MO, USA). The 4′,6-diamidino-2-phenylindole, dihydrochloride (DAPI) solution was purchased from Thermo Fischer Scientific (Waltham, MA, USA). Cell culture dishes were from Corning Incorporated (Corning, NY, USA), Greiner (Kremsmünster, Austria), or Nunc (Roskilde, Denmark).

### 2.2. Syntheses

#### 2.2.1. PAMAM G2 and G3 Substituted with Glycidol

PAMAM G2 and G3 dendrimers were derivatized with 25% molar excess of *rac*-glycidol (gl), *R*-gl, or *S*-gl in relation to twice the number of dendrimer primary amine groups. In a typical procedure, to 120 mg G3 (17.3 µmoles) in 2 mL methanol, neat gl was added dropwise (92 µL, 1.4 mmol) with vigorous stirring. The mixture was stored for eight hours at room temperature, transferred into a cellulose dialytic bag (MW_cutoff_ = 1 kD for G2 and 3.5 kDa for G3), and dialyzed for three days against water (7 × 2.5 L). Then, the solvent was removed by vacuum rotary evaporation and the oily residue dried under reduced pressure overnight (<2 mbar). The products were characterized by ^1^H NMR spectroscopy as fully derivatized **G2^32*Rac*gl^**, **G2^32*R*gl^**, **G2^32*S*gl^**, **G3^64*Rac*gl^**, **G2^64*R*gl^**, and **G2^32*S*gl^**. The isolated yield was > 90% in every case.

In a series of synthesis, the deficient amount of *S*-gl and *R*-gl was used to convert G2 into **G2^9*S*gl^**, **G2^14*R*gl^**, **G2^22*S*gl^**, and **G2^22*R*gl^** by using the same protocol as above except the number of gl equivalents in the reaction mixture corresponded to 25% of dendrimer primary amine groups for the first two derivatives and 75% of available amine groups of G2 in two others. The products were purified as before. The partially substituted derivatives were characterized by ^1^H, ^13^C, and 2-D correlations spectroscopy (COSY), heteronuclear single quantum correlation (HSQC), and heteronuclear multiple bond correlation (HMBC) spectroscopy. Then, the **G2^9*S*gl^** and **G2^22*S*gl^** were further reacted with *R-*gl to complete bis-substitution of every G2 amine group with gl residues, while **G2^22*R*gl^** and **G2^14*R*gl^** were fully substituted with *S*-gl. In both cases, 20% molar excess of gl in relation to the available number of amine equivalents of the substrate was used. The resulting species were identified by NMR spectroscopy and the mixed-enantiomeric fully substituted G2^gl^ spectra were compared with **G2^32*Rac*gl^** spectra.

#### 2.2.2. PAMAM G2 and G3 Single-Labeled with Fluorescein Isothiocyanate and Rhodamine B Isothiocyanate

The G2 and G3 dendrimers were single labeled with fluorescein isothiocyanate (FITC) and rhodamine B isothiocyanate (RBTC) by stepwise addition of one equivalent of 10 mM ethanolic solution of FITC (8 and 4 mL) into 4 or 2 mL of 20 mM methanolic G2 or G3 solution (80 and 40 µmoles, respectively). Labeling with RBTC was performed by addition of solid RBDC into 10 mM solution of G2 or G3 in methanol. Four products (G2^F^, G2^R^, G3^F^, and G2^R^) were isolated as solids, which were since protected from daylight, dissolved in 6 mL methanol, divided into two equal portions, and further converted by reaction with *S*-gl or *R*-gl used in 25% molar excess. The obtained eight products were not fully substituted with gl residues and identified by ^1^H NMR spectroscopy as **G2^16*R*glF^**, **G2^19*S*glF^**, **G3^19*R*glF^**, **G3^35*S*glF^**, **G2^19*R*glR^**, **G2^30*S*glR^**, **G3^45*R*glR^**, and **G3^35*S*glR^**. These fluorescent-labeled compounds were used for biological studies.

### 2.3. NMR Spectroscopy

The 1-D ^1^H and ^13^C NMR spectra and 2-D ^1^H-^1^H correlations spectroscopy (COSY), ^1^H-^13^C heteronuclear single quantum correlation (HSQC), and heteronuclear multiple bond correlation (HMBC) spectra were recorded in deuterated water using Bruker 300 MHz (Rheinstetten, Germany) and worked up with TopSpin 3, 5 software at College of Natural Sciences, University of Rzeszów.

### 2.4. Biological Methods

#### 2.4.1. Cell Culture

Human immortalized keratinocytes (HaCaT) were grown in DMEM and human squamous carcinoma cells SCC-15 (CRL-1623 ATCC) were cultured in DMEM/F-12 supplemented with hydrocortisone (400 ng/mL). Culture media were supplemented with heat-inactivated 10% FBS and 100 U/mL penicillin and 1% streptomycin solution. Both cell lines were cultured at 37 °C in a humidified atmosphere with 5% CO_2_ with media changed every 2–3 days. Cells were passaged at 70–80% confluence after trypsinization with 0.25% trypsin–EDTA in calcium- and magnesium-free PBS. Cell morphology was monitored using Nikon TE2000S Inverted Microscope (Tokyo, Japan) with phase contrast. The number and viability of cells were estimated by the trypan blue exclusion test with Automatic Cell Counter TC20TM (BioRad Laboratories, Hercules, CA, USA). All assays were performed in triplicates in three independent experiments.

#### 2.4.2. Toxicity Assay

HaCaT and SCC-15 cells were seeded in flat-bottom 96-well culture plates in triplicate (100 µL cell suspension per well) at a density of 1 × 10^4^ cells/well and allowed to attach for 24 h. The stock solution of 3 mM dendrimers was filtered with sterile syringe filters (0.22 µm) and used to treatment of cells with a range of increasing concentrations from 0 to 300 µM (100 µL/well) for 24 h in 37 °C. After that, the neutral red assay was performed as described [24].

#### 2.4.3. Cellular Internalization of Dendrimers

The HaCaT and SCC-15 cells were seeded into, black, 96-well microtiter plates at a density of 4 × 10^4^ cells/well and placed in an incubator for 24 h. Next, cells were incubated with 1 µM solutions of FITC-labeled dendrimers (dissolved in culture medium) for 0.1, 0.5, 1, 2, 3, or 4 h. Then, dendrimer solutions were removed and plates were washed three times with PBS to remove unbound dendrimers. Fluorescence was read at 485 nm/530 nm (exc./em.) with Infinite M200 PRO Multimode Microplate Reader (TECAN Group Ltd., Maennedorf, Switzerland). The median of the triplicate sample measurements was calculated after the background values (cells alone) were subtracted. The fluorescence intensity was calibrated using FITC-labeled dendrimers diluted in water.

#### 2.4.4. Visualization with Confocal Microscopy

HaCaT and SCC-15 cells were seeded at a density of 7 × 10^4^ cells/well in microscopic, eight-chamber slides with an ultra-thin bottom (Nunc, Roskilde, Denmark) for 48 h in 400 μL of complete medium. FITC- or rhodamine B -labeled *S*- and *R*-glycidol furnished dendrimers at 100 nM concentrations were added (300 μL/well). After four hours of incubation and washing three times with PBS, the cells were fixed with 3.7% formaldehyde for 10 min and stained with 600 nM DAPI solution in PBS (15 min at room temperature). Images were collected with a confocal microscope (Olympus FV10i, Tokyo, Japan) at 488/530 nm for FITC, 556/573 nm for rhodamine B, and 405/461 nm for DAPI. Images were collected using an objective with water immersion, under a magnification of 180× in the Z-axis position at the largest nuclear cross-section area. The pinhole was set for 1 AU (airy unit), and the obtained images had an optical section thickness of approximately 1.02 μm. The laser power and sensitivity were constant. Raw images were collected and saved in oif format, and the processing was carried out in ImageJ.

To visualize the concentration-dependent accumulation of *S*- and *R*-glycidol furnished dendrimers, HaCaT and SCC-15 cells were seeded in ultra-thin, flat-bottom, 96-black well culture plates (1 × 10^4^ cells/well) and incubated for 24 h. Afterward, working solutions of FITC labeled dendrimers in the range of 0–300 µM concentration were added, incubated for 24 h, and washed three times with PBS. Samples were observed and the images were collected with Olympus IX-83 fluorescent microscope (Tokyo, Japan) using an objective with 20 × magnification. The light intensity and sensitivity were constant.

#### 2.4.5. Statistical Analysis

To estimate the differences between treated and non-treated control samples, statistical analysis was performed using the non-parametric Kruskal–Wallis test due to the lack of a normal distribution of data in the studied groups (analysis with Shapiro–Wilk test). Comparisons of *R* and *S* glycidol furnished dendrimers were assessed with Mann–Whitney U test. *p* < 0.05 was considered statistically significant. Calculations were performed using Statistica 13.3 software (StatSoft, Tulsa, OK, USA).

## 3. Results and Discussion

Polyamidoamine (PAMAM) dendrimers are well soluble in water. They are currently tested in many laboratories as drug carriers. However, high full-generation PAMAMs are hemotoxic [25] due to surface primary amine groups, which make them strongly basic. Therefore, amine groups must be partially derivatized if the drug-PAMAM conjugate is designed. Convenient conversion of amine group maintaining dendrimer water solubility is polyhydroxylation with lactones, which convert dendrimer primary amine into amide groups or with glycidol (gl), which adds in stoichiometry 2:1 to a primary amine. Although racemic gl was often used as an amine blocking group, no detailed studies on optically pure *R*-gl or *S*-gl derivatives of PAMAM dendrimers were reported till now. Since gl covered PAMAM dendrimers are often used as macromolecular anticancer drug carriers, this inspired us to test the interaction of these derivatives with human normal and cancer cells.

The products of the reaction between gl and PAMAM G1 were thoroughly characterized by NMR spectroscopy, polyacrylamide gel electrophoresis, capillary electrophoresis, MALDI-TOF and ESI mass spectrometry, and acid-base potentiometric titration, together with pristine PAMAM G1 and peracetylated derivative in order to detect and quantitatively determine skeletal and substitutional imperfections [22]. We restudied the products of the reaction between PAMAM G2 and G3 and both enantiomers of gl to simplify the NMR spectral pictures and also to characterize the species with a deficient number of gl substituents. The synthetic routes and average stoichiometry of obtained compounds are presented in Scheme 1, together with atom numbering of atoms for spectral assignment. The NMR spectroscopy was used to identify all products as follows.

### 3.1. NMR Spectroscopy Studies

#### 3.1.1. Comparison of the ^1^H and ^13^C NMR Spectra of PAMAM G2 and S-Glycidol Covered PAMAM, G2*^S^*^gl^

^1^H NMR spectroscopy enables quantifying the level of PAMAM dendrimer substitution with glycidol. The ^1^H and ^13^C NMR spectra of substrate PAMAM **G2** and **G2*^S^*^gl^** are shown in Figure 1 and Figure 2, respectively. Only one proton resonance for all protons b (b_2_, b_1_, and b_0_) was observed at 2.42 ppm and used as an internal integral reference of intensity [56H] for **G2** and **G2*^S^*^gl^** (for C and H atoms numbering, see Scheme 1). Both the chemical shift and intensity of the triplet remained unaltered throughout the series of all neat and gl-modified dendrimers. The vicinal protons triplets of a_2_a_1_a_0_ at 2.82 also remained unaltered.

The triplets of protons c_2_ and d_2_ localized in outer shell (shell 2 in case of **G2**) were observed at 3.23 and 2.71 ppm, respectively, with [32H] integral intensity both. Inner sphere c and d resonances showed overlapping triplet from c_1_,c_0_ at 3.29 ppm (intensity [24H]) and d_1_,d_0_ (overlapped with core singlet of intensity [4H], totally [28H]) for **G2** (Figure 1A). Upon addition of gl into terminal amine groups, the outer sphere c_2_ resonance shifted into 3.3. ppm-centered multiplet and overlapped with c_1_c_0_ triplets within the 3.32–3.38 region. The total intensity of c_2_c_1_c_0_ resonances corresponds to [56H]. The resonance from d_2_ is spread over a broad 2.75–2.57 region and overlapped with d_1_, d_0_, and o from **G2** core and terminal nitrogen-attached methylene group of glycidol. Additionally, resonances from the latter show multiplet structure due to the presence of diastereotopic -CH_2_- next to the chiral center at carbon f.

The ^1^H *S*-gl resonances of protons g and f of integral intensity corresponding to [32H] and [64H] are unresolved multiplets and AB-type spectrum of gg’ diatereotopic pair, respectively (Figure 1B).

The aliphatic regions of ^13^ C NMR spectra of **G2** and **G2*^S^*^gl^** are shown in Figure 2A,B, respectively. The ^13^C resonances were assigned based upon standard HSQC and HMBC experiments (the combined HSQC and HMBC spectrum of **G2*^S^*^gl^** is presented in Appendix A). Thus, the considerable shift of outer-sphere d_2_ (−4.1 ppm) and c_2_ (+14.5 ppm) resonances occur upon substitution of terminal NH_2_ groups with two *S*-gl substituents, with d_1_d_0_ and c_1_c_0_ resonances almost unaltered. Three ^13^C resonances of *S*-gl substituents indicate that all terminal amine groups were uniformly substituted with two gl-*S* each. Analogous ^1^H and ^13^C NMR spectra were obtained for **G2*^R^*^gl^**. Similar ^1^H and ^13^C NMR spectral pattern was obtained for higher generation PAMAM dendrimers, namely, G3, G4, and G5, which were substituted with 64, 128, and 256 enantiomerically pure *R*- or *S*-gl, respectively. In comparison with the spectra of **G2*^S^*^gl^** or **G2*^R^*^gl^**, the only differences observed were due to the integral intensity of resonances.

The simplicity of ^13^C NMR spectra of PAMAM G2 and G3 dendrimers substituted with enantiomerically pure *S*-gl or *R*-gl, which show only one set of gl resonances for **G2^32*S*gl^** or **G2^32*R*gl^** and corresponding fully converted G3–G5 (spectra not shown) indicates that enantiomeric purity of *S*-gl is retained in conversion to formally *R*-chiral *n*-2(*R*), 3-dihydroxypropyl substituents. Unfortunately, the chirality checked with CD showed the Cotton effect on the almost background level, as it was observed also for other chiral dendrimers of radial symmetry [16]. In most reactions, the enantiomerically pure glycidols retain the C-2 configuration such as, for instance, the esterification with primary and secondary alcohols leading to 1-O-alkyl-*sn*-glycerol [26]. However, the pure inversion of configuration has also been observed in conversion of gl with diethylaluminum cyanide leading to 1-cyano-2,3-diols [27].

Although we have no evidence to claim retention or inversion of gl configuration after addition to dendrimer primary amine of G2, the simplicity of NMR spectra of **G2*^S^*^gl^** and **G2*^R^*^gl^** indicate that homochiral dendrimers were obtained in contrary to quite complex ^13^C NMR spectra of *rac*-gl derived dendrimers (vide infra).

#### 3.1.2. Deficient Substitution of PAMAM G2 Dendrimer with Optically Pure Glycidol

In the substoichiometric conversion of PAMAM G2 with *S*- or *R*-glycidol, the species with double and single-substituted terminal amine groups were obtained and characterized by NMR spectroscopy. The ^1^H NMR spectra enabled us to determine the level of terminal amine group substitution with 2,3-dihydoxyalkyl substituents added in the reaction of G2 with enantiomerically pure glycidol (Figure 3).

Stepwise addition of *S*-glycidol resulted in an increase of proton g multiplets intensity (within 3.45–3.65 ppm) region related to internal reference triplet of a_2_a_1_a_0_ triplets of overall intensity [56H]. Three G2^ngl*S*^ species were obtained, where average *S*gl molecules added (*n*) was 9, 14, and 22 (see Figure 3, spectra B, C, and D). The AB-type spectrum of terminal g protons of 2, 3-dihydroxypropyl substituent in **G2^32*S*gl^** (Figure 3E) is vicinally coupled with proton f. Quartets of g and g’ protons are centered at 3.61 and 3.50 (δ_AB_ = 0.11 ppm) with geminal coupling constant *J*_gem_ = 11.7 Hz and different vicinal coupling constants *J*(gf) = 4.2 Hz and *J*(g’f) = 5.9 Hz.

When an under-stoichiometric amount of *S*-gl was used the **G2^9*S*gl^** was obtained, containing some unsubstituted amine groups, mostly mono-substituted amine groups with 2, 3-dihydroxypropyl, and a minor amount of double substituted amine groups. The AB-type spectrum of gg’ protons on single substituted amine groups showed slightly lower magnetic nonequivalence (5.59 and 3.50 ppm, δ_AB_ = 0.11 ppm), coupled geminally with the same *J*_gem_ = 11.7 Hz, while vicinal constants were larger than for double-substituted arms, namely, *J*(gf) = 4.6 Hz and *J*(g’f) = 6.7 Hz (see Figure 3, trace B). In the series of higher substituted G2, namely **G2^14*S*gl^** and **G2^22*S*gl^** the AB spectra of gg’ protons overlapped and no reliable deconvolution of multiplets could be performed in order to quantify the number of unsubstituted, mono- and bis-substituted arms of G2. The additional ^1^H NMR changes accompanied a stepwise conversion of G2. Single and double substitution of terminal amine group resulted in transformation of primary amine into secondary and eventually tertiary terminal amine group. The chemical shifts of protons c_2_ and d_2_ changed due to this transformation. In the case of **G2**, c_2_ proton resonance was initially observed as a triplet at 3.24 ppm (Figure 3, trace A), together with c_1_ and c_0_ at 3.30 ppm, while the stepwise addition of gl resulted in the decrease of 3.24 ppm triplet, which eventually disappeared in the spectrum of fully substituted **G2^32*S*gl^** (Figure 3, trace E) and was replaced by multiplet centered at 3.30 ppm of intensity [56H], corresponding to all c protons of **G2^32*S*gl^**. Similar changes occur in the region of d_2_ proton resonances; however, the multiplets of all d protons overlap with e proton resonances of attached glycidol. The detailed changes could be further monitored using ^13^C NMR spectra and heteronuclear ^1^H–^13^C NMR experiments (vide infra).

The aliphatic regions of ^13^C NMR spectra of G2^ngl*S*^ (where *n* = 0, 9, 14, 22, and 32) are shown in Figure 4. The ^13^C resonances were assigned based upon standard HSQC and HMBC experiments. We found that PAMAM inner shell resonances remained unaffected upon terminal amine group substitution, while d_2_ and c_2_ proton resonances shifted from 42.3 and 40.5 in amine-terminated G2 into 39.3 and 48.1 upon mono- and 37.7 and 54.2 ppm upon disubstitution with gl*S*, respectively. The resonances of *S*-glycidol-derived 2(*R*), 3-dihydroxypropyl substituents for single-substituted arms are located at 51.3, 64.7, and 71.0 ppm for e, f, and g carbon nuclei (labeled as *e^S^*, *f^S^*, and *g^S^*, in blue color in Figure 4), while the appropriate resonances for double-substituted analogs are observed at 57.6, 64.6, and 69.8 (labeled as *e^SS^*, *f^SS^*, and *g^SS^* in red color in Figure 4). Thus, the largest chemical shift differences between single- and double-substituted arms are those in closest e, d, and c carbon nuclei, as one could expect. It should be also noticed that even in the ^13^C NMR spectrum of **G2^22*S*gl^** the resonances from non-substituted arms are clearly visible. This corresponds to the average number of six double-substituted arms, 10 mono-substituted, and 16 unsubstituted arms if assume that the intensities of carbon-13 resonances *d_2_^SS^*, *d_2_^S^*, and d_2_ are nearly equal.

The analogous result was obtained using *R*-gl stepwise addition to G1, G3, and G4 PAMAM dendrimers. The series of three ^1^H NMR spectra for **G2^15*R*gl^**, **G2^22*R*gl^**, and **G2^32*R*gl^** are shown in Appendix A.

#### 3.1.3. Fluorescein and Rhodamine B Single-Labeled Homochiral Dendrimers

The fluorescent labels, FITC and RBTC were carefully attached to G2 and G3 dendrimers. Single-labeled G2^F^, G2^R^, G3^F^, and G2^R^ were then converted into *n*-(2, 3-dihydroxypropyl) derivatives with the same protocol, using 20% molar excess of enantiomerically pure *S*- and *R*-gl in relation to available amine groups (two equivalents per one terminal amine group). Obtained derivatives were not fully substituted. We obtained series of derivatives, namely, **G2^16*R*glF^**, **G2^19*S*glF^**, **G3^19*R*glF^**, **G3^35*S*glF^**, **G2^19*R*glR^**, **G2^30*S*glR^**, **G3^45*R*glR^**, and **G3^35*S*glR^**. The average stoichiometry of compounds was determined by ^1^H NMR spectroscopy (spectra not shown). These compounds were then tested as single dendrimers and in homochiral pairs labeled with two different fluorescent labels on human cell cultures (vide infra).

#### 3.1.4. Products of Stepwise Addition of Enantiomerically Pure gl into G2 Dendrimer

Fully gl-substituted PAMAM dendrimers require a temperature regime to avoid polyhydroxylation, which starts already at a temperature above 50 °C. When 20% molar excess of gl is used, the number of glycidol residues attached to PAMAM becomes higher than twice the number of terminal amine groups of PAMAM dendrimer as determined by the integral intensity of glycidol-derived multiplets of protons f. On the other hand, the substoichiometric amount of gl results in the formation of dendrimers with double- and single-substituted arms. Thus, we could obtain **G2^9*S*gl^** and **G2^22*S*gl^** and characterize them by NMR spectroscopy (vide supra).

These two species were then used to convert remaining free amine groups and mono-substituted ones into bis-substituted ones. Thus, we obtained the average substituted **G2^9*S*gl23*R*gl^** and **G2^22*S*gl10*R*gl^**. The corresponding ^1^H NMR spectra enabled us to identify these species as fully substituted, although they did not illustrate the symmetry of mixed-isomer derivatives. Therefore, we examined the ^13^C NMR, which are presented for starting **G2^9*S*gl^** and **G2^22*S*gl^** (Figure 1A,E) and mixed **G2^9*S*gl23*R*gl^** and **G2^22*S*gl10*R*gl^** (traces B and D, respectively). There are essentially two sets of gl resonances observed in the ^13^C NMR spectra of the products, which were assigned to the chiral arm and *meso* arm. Two sets of carbon f resonances can be identified at ca 70 ppm (downfield shifted *f^SS^* and *f^RR^* for chiral arm and upfield shifted *f^SR^* for *meso* arm (Appendix A, traces B and D). However, even in the spectra of compounds **G2^9*S*gl23*R*gl^** and **G2^22*S*gl10*R*gl^**, which were obtained under kinetic control, there are some other species of highly distorted symmetry, which show low-intensity f carbon resonances shifted downfield and g carbon resonance shifted upfield. These signals are much pronounced in the spectrum of **G2^32*Rac*gl^** (trace C), which is obtained by one portion addition of *rac*-gl into G2 dendrimer. These puzzling symmetry arms might origin from elongated polyhydroxy substituents because on the one hand the intensity of gl ^1^H NMR resonances clearly indicates that the number of attached gl residues is 32; on the other hand, in the ^13^C NMR spectrum, the resonance from the single-substituted arm is still present at ca 51 ppm (Appendix A, trace C). It should be emphasized that this spectral pattern is reproducible for reported G1^16*Rac*gl^ [20] and for fully substituted G3^64*Rac*gl^, G4^128*Rac*gl^, and G5^256*Rac*gl^ (not shown here).

### 3.2. Biological Studies

#### 3.2.1. Dendrimers Toxicity

To perform cytotoxic studies, we chose two human cell lines—immortalized keratinocytes (HaCaT) and squamous carcinoma cells (SCC-15)— both belonging to epithelial tissue. SCC-15 is a model of cancer cells used as a target for the synthesized PAMAM conjugates occurring as a drug delivery system, while HaCaTs comparatively as non-tumorigenic cells. We used immortalized, non-tumorigenic keratinocytes, compared to normal ones, since the latter from skin biopsies indicate high variations between passages, short culture lifetime, and donor variability [28]. To assess the cytotoxicity, a neutral red assay was performed as one of the most sensitive viability tests [29].

This assay indicated, that G2 and G3 PAMAM dendrimers furnished with enantiomers *S* or *R* glycidol (for simplicity of notation the number of gl residues will be omitted in the abbreviations of homochiral derivatives, which will be further named as **G3*^R^*^glF^, G3*^S^*^glF^, G2*^R^*^glF,^** and **G2*^S^*^glF^**) were highly biocompatible and did not indicate cytotoxic action against both immortalized human keratinocytes (HaCaT) and human squamous carcinoma cells (SCC-15) up to 300 µM concentration (Figure 5).

Only at the highest 300 µM concentration of **G3*^R^*^glF^** and **G2*^S^*^glF^** cancer cell viability increased by about 20 and 35%, respectively. Obtained results show the increased biocompatibility of PAMAM dendrimers after coating with glycidol residues. In our earlier studies, we estimated that native G3 PAMAM dendrimers after 24 h incubation evoked significantly higher cytotoxicity against SCC-15 and HaCaT cell from 10 and 50 µM concentrations, respectively [28,30]. The studied compound had also lower toxicity than dendrimers substituted with hydroxyl groups *D*-glucoheptono-1,4-lactone [12]. Therefore, these compounds meet the requirements for drug carrier, i.e., high biocompatibility and high water solubility [31]. Compared to other PAMAM dendrimers, PAMAM–pyrrolidone dendrimers, and poly (propylene imine) dendrimers, the result should be considered highly satisfactory [32]. For example, considered to be low-toxic PAMAM G4, pyrrolidone dendrimer was toxic already after two or six hours incubation with 200 µM concentrations against mouse neuroblastoma (N2a) cells (viability lowering for about 40%) [33]. A 25% decrease in viability of mHippoE-18 embryonic mouse hippocampal cells was also detected after 200 µM concentration treatment [34].

#### 3.2.2. Time-Dependent Cellular Accumulation

Another requirement for drug delivery systems (DDS) is targeting specific sites or cell populations [31,35]. In anticancer therapy, selective action, uptake, and accumulation of DDS in cancer cells are very important since they can diminish adverse side effects against normal cells and tissues [36]. Therefore, we tested homochiral dendrimer conjugates interaction with human cancer or non-tumorigenic cell cultures to find whether they can be recognized by cell membrane receptors and selectively accumulated in cancer cells.

It has been observed before, that gel-entrapped lipase from *Rhizopus javanicus* enantioselectively catalyzed esterification of R-(+)-gl with *n*-butyric acid. However, the highest enantioselectivity was achieved after pretreatment of silica gel templated lipase with R-(-)-2-octanol [37].

The enzymes catalyzing water addition to epoxide with concomitant ring-opening show a variety of stereoselectivity of diol products, with retention or inversion of configuration, depending on substituents on gl and enzyme origin [38]. The involvement of lipases and/or epoxidases in the chiral recognition of homochiral dendrimers is inevitable if they reach cell membrane, cytosol, and other organelles.

The 2,3-hydroxypropyl substituents are formed upon food treatment, they covalently attach into Val and were found as N-2, 3-dihydroxypropyl Val residues in hemoglobin. This derivative was considered as a marker in food examination [39]. Neither in this toxicological study nor in other cases [40] the enantiomerically pure gl derivatives were studied separately.

Our fluorimetric studies with fluorescently labeled enantiomers of dendrimers revealed, that in both tested cell lines, *R-*gl enantiomer-derived G3 dendrimer conjugates (**G3*^R^*^glF^**) were always significantly stronger accumulated than *S-*gl enantiomer-derived ones (**G3*^R^*^glF^**) after 0.1–4 h incubation with 1µM concentrations. An analogous effect was observed also in the case of G2 dendrimers, but only in cancer cells and from already 1 h incubation (Figure 6A). Additionally, cancer cells accumulated all studied compounds much efficiently than keratinocytes (on average 67%) (Figure 6A).

Obtained data are in accordance with confocal microscopy images (Figure 6B). Regardless of the fluorescent marker used, there is a greater accumulation of *R* forms of the enantiomers than the *S*. Conjugates of G3 dendrimer are taken up, transported, and stored mainly in cellular vesicles (mainly lysosomes), while G2 conjugates are much more scattered in the area of the cytoplasm, with visible slightly lower accumulation in intracellular vesicles (Figure 6B). In addition, enantiomers based on G2 dendrimers show a higher degree of penetration into the cell nuclei and the ability to bind to the nucleoli of cells (pointed by yellow arrows). Differences in the distribution of compounds in various areas of cells will significantly affect the possibility of using them as carriers of specific drugs, depending on the destination area [41].

It was also observed that the location of *R*-gl- and *S*-gl-covered G3 dendrimers in intracellular space was similar. Colocalization of FITC and rhodamine B labeled enantiomers was high in both cell lines (Figure 6B); therefore, there is no evidence that the intracellular transport of both enantiomers differs significantly. Only a dendrimer generation seems to be a factor influencing the distribution of dendrimers in the cell. It was previously proved, that –OH terminated PAMAM dendrimers are transported mainly via clathrin-dependent pathway and macropinocytosis [41] and through caveolin- and clathrin-independent pathway [42], depending on the type of cells tested.

Figure 6B shows that G3^gl^ dendrimers indicated a high affinity for cell membranes, especially SCC-15 tumor cells, as indicated by white arrows. A similar effect was observed by Albertazzi et al. in HeLa cells after incubation with fluorescently labeled G4 PAMAM dendrimer [41]. The observed phenomenon was weaker in HaCaT cells, and these differences may be connected with variations in transfer efficiency of studied nanoparticles into cells. To visualize concentration-dependent differences in the accumulation of dendrimers in both cell lines, we also performed additional visualization in a fluorescence microscope (Figure 7). Cells were incubated with conjugates **G3*^R^*^glF^, G3*^S^*^glF^, G2*^R^*^glF,^** and **G2*^S^*^glF^** in the range of 0 –300 µM concentrations for 24 h. Images illustrate that the number of dendrimers in the cells increased with their concentration and regardless of the concentration all compounds were more efficiently taken up and accumulated in SCC-15 tumor cells than in keratinocytes (Figure 7).

## 4. Conclusions

PAMAM G2 and G3 dendrimers were quantitatively converted by the addition of enantiomerically pure *R*-glycidol and *S*-glycidol into primary amine groups to obtain bis-(2, 3-dihydroxypropyl) derivatives. Fully substituted homochiral derivatives had effective radial symmetry, which was illustrated by one set of 2, 3-dihydroxypropyl carbon resonances in ^13^C NMR spectra. Deficient substitution of G2 or G3 with enantiomerically pure glycidol enabled us to identify the mono-substituted arms of dendrimer along with bis-substituted ones by ^13^C NMR spectroscopy. The single-fluorescent labeled homochiral G2 and G3 dendrimers were obtained in order to test their ability to bind with the cell membrane and enter into different human cells in vitro.

Biological studies with human non-tumorigenic keratinocytes (HaCaT) and squamous carcinoma cells (SCC-15) indicated that *S*- and *R*-glycidol furnished dendrimers were highly biocompatible with no toxicity up to 300 µM concentrations in contrast to the pristine PAMAM dendrimers, for which significant toxicity was observed already within 10–50 µM concentrations. The *R* and *S* glycidol derivatives of G3 PAMAM dendrimer bind to the cellular membrane and enter into the cells with higher accumulation in cancer cells. The *R* enantiomer conjugate quantity in studied cells was always significantly higher than *S* ones. The G3^gl^ dendrimers were taken up more efficiently than G2^gl^ derivatives; however, G2^gl^ derivatives penetrated also nuclei of studied cells.

## Data Availability

The data presented in this study are available on request from the corresponding author.

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
