# Peer review of "Chiral Recognition of Homochiral Poly (amidoamine) Dendrimers Substituted with R- and S-Glycidol by Keratinocyte (HaCaT) and Squamous Carcinoma (SCC-15) Cells In Vitro"

_polymers, 2021, doi:10.3390/polym13071049_

Round 1

Reviewer 1 Report

The manuscript entitled "Chiral recognition of homochiral PAMAM dendrimers substituted with R- and S-glycidol by keratinocyte (HaCaT) and squamous carcinoma (SCC 15) cells in vitro" modified G2 and G3-PAMAM dendrimers with different enantiomers and evaluated the interaction between materials and cells. The characterization of materials were carefully analyzed by NMR. The viability of cells were studied to investigate the material toxicity. Using flourescent probes, materials were investigated their accumulation on cell surface or inside cells. However, there were still some problems need to be revised before acceptance, such as:

  1. In the first paragraph, could you mention in detail your main opinions instead of “… although combinations of chiral centers allow to specific other chiral dendrimers as it was discussed and reviewed [14].”?
  2. Please explain “ee” in the 58th line before using abbreviation.
  3. Be similar to my comment 1, could you give your opinion in brief instead of writing “and described these phenomena as follows” (line 113)?
  4. In your introduction, could you cite more related papers? such as: “PEGylated PAMAM dendrimers loading oxaliplatin with prolonged release and high payload without burst effect”, Biopolymers, 110 (7), e23272; “Recent progress and advances of multi-stimuli-responsive dendrimers in drug delivery for cancer treatment”, Pharmaceutics 11 (11), 591; Starburst pamam dendrimers: Synthetic approaches, surface modifications, and biomedical applications, Arabian Journal of Chemistry, Volume 13, Issue 7, July 2020, Pages 6009-6039; …
  5. In line 137, “stored” or “reacted” ?
  6. In 2.4.4, could you add exactly the unit after cell number?
  7. In line 246, “TOF”, not “TofF”. In line 132, please correct “syntheses”
  8. In the figure 5, Y name “vibility” should be corrected into “viability”
  9. Could you add the chemical struture in the corresponding NMR for easy checking? Please write exactly 1H- or 13C-NMR instead of H- or C-NMR (exp: figure caption,…)
  10. How can you determine the amine residue of G2 or G3 after modification? Please double check with quantitative determination of amine.
  11. In case of comparison between PAMAM-pyrrolidone and other ones, could you summarize the results with citations, not just write “… highly satisfactory (see ref [30]”?
  12. How about the molecular weight of G2, G3 and modified G2, modified G3? You should operate GPC for your polymer to strongly confirm your synthesized polymer beside NMR.

Author Response

REVIEWER 1

The manuscript entitled "Chiral recognition of homochiral PAMAM dendrimers substituted with R- and S-glycidol by keratinocyte (HaCaT) and squamous carcinoma (SCC 15) cells in vitro" modified G2 and G3-PAMAM dendrimers with different enantiomers and evaluated the interaction between materials and cells. The characterization of materials were carefully analyzed by NMR. The viability of cells were studied to investigate the material toxicity. Using flourescent probes, materials were investigated their accumulation on cell surface or inside cells. However, there were still some problems need to be revised before acceptance, such as:

  1. In the first paragraph, could you mention in detail your main opinions instead of “… although combinations of chiral centers allow to specific other chiral dendrimers as it was discussed and reviewed [14].”?

ANSWER:

In the paper cited as [14] the chiral dendrimer concept and cases are nicely reviewed and encountered. We have left only the citation, not going too far into details because of limited space in our INTRODUCTION. The Reviewer 2 encouraged us to shorten both Introduction and Conclusion paragraphs. In our work we focused on surface modification with chiral substituents on surface (class 3 homochiral dendrimers in [14])

We have removed the fragment on another class of homochiral dendrimers specified in [14], leaving only first part of the sentence, which is not the matter of our opinion, just the facts.

AMENDMENT:

The short sentence is now in lines 54-55; changes are highlighted in blue:

Generally the chirality in dendritic architecture is achievable using chiral core and build-up of dendritic arms (class 1), using chiral building blocks (class 2), and encountered above end-group chiral substituents (class 3) [16].

  1. Please explain “ee” in the 58th line before using abbreviation.

ANSWER:

Although ee is widely known parameter for asymmetric catalysis, we realize that readers of Polymers might not be familiar with enantiomeric excess.

AMENDMENT:

We have given full name of ee in the sentence in lines 62-63, which is now:

…..chemoselectivity and > 90% enantiomeric excess (ee) in presence of catalytically active Ru(I) and Rh(II) organometallic complexes [17].

  1. Be similar to my comment 1, could you give your opinion in brief instead of writing “and described these phenomena as follows” (line 113)?

ANSWER:

We meant that our biological observations will be described in following sections. Probably my language did not follow my intention. Therefore we have changed it according to suggestion into short description what has been done in this paper.

AMENDMENT (corrected sentence, amendment in blue, lines 109-111 in the revised version):

Surprisingly we have observed some enantiodiscrimination of homochiral dendrimers in cell membrane interaction by both types of cells and generation dependent internalization.

  1. In your introduction, could you cite more related papers? such as: “PEGylated PAMAM dendrimers loading oxaliplatin with prolonged release and high payload without burst effect”, Biopolymers, 110 (7), e23272; “Recent progress and advances of multi-stimuli-responsive dendrimers in drug delivery for cancer treatment”, Pharmaceutics 11 (11), 591; Starburst pamam dendrimers: Synthetic approaches, surface modifications, and biomedical applications, Arabian Journal of Chemistry, Volume 13, Issue 7, July 2020, Pages 6009-6039; …

ANSWER:

Thank you very much for suggestions. We knew already all these crucial papers and keep them on file. The only reason we did not decide to mention PEG-ylation as method of dendrimer modification was that there is so many applications of PEG-ylated PAMAM dendrimers that we considered PEG-ylation as obvious way to eradicate “toxic” amine groups of PAMAM. Especially we consider importantance of the oxaplatinum encapsulates, because one of us just started his chemistry with Pt(II) and Pd(II) model compounds and interaction with peptide and nucleotides (SW in 1979).

This was our shortcoming. We have cited now two of suggested papers:

AMENDMENT:

The recent reviews on dendrimers have been added in the Introduction, sentence in lines 49-52 in the revised version (added text in blue):

Thus PAMAM chiral dendrimers can be obtained that are surface-modified with various substituents, derived from D-gluconolactone [10], D-glucoheptono-1,4-lactone [6,12], aminoacids [13], and by PEG-ylation [14, 15].

  1. In line 137, “stored” or “reacted” ?

ANSWER: Yes, technically we stored the mixture at room temperature. The addition of glycidol into amine groups goes on at room temperature. This is an important factor of procedure, because at elevated temperature (above 50 C degrees) the glycidol reacts also with hydroxyl groups of 2,3-dihydroxypropyl groups derived from already added glycidol, which results in propagation of polyhydroxyalkyl chain on amine groups. Also the important factor is low excess of added glycidol, which should be kept below 25 molar excess %; otherwise polyhydroxyalkylation is involved also. It took us some time to adjust reaction conditions.

Thus, the statement “stored” instead of reacted is more proper, because there are other reactions which go on all the time, even if the conditions are held as given in the protocol.

Please, agree to leave the statement as such.

  1. In 2.4.4, could you add exactly the unit after cell number?

AMENDMENT:

The unit was added (line 209 in the revised version).

  1. In line 246, “TOF”, not “TofF”. In line 132, please correct “syntheses”

ANSWER:

Thank you, we amended this abbreviation; the old abbreviation of Time of Flying was TofF.

AMENDMENT:

Line 250 (in blue in the revised version)

  1. In the figure 5, Y name “vibility” should be corrected into “viability”

ANSWER:

Thank you very much. We missed this typo during spell checking of main text.

We have corrected Figure 5.

  1. Could you add the chemical struture in the corresponding NMR for easy checking? Please write exactly 1H- or 13C-NMR instead of H- or C-NMR (exp: figure caption,…)

In original version the formula was given at Scheme 1.

ANSWER:

Yes, we had only doubts, where to place the formula, because all spectra need formula close to the spectrum to follow the spectral details. Considering 1-H and 13-C; yes, these abbreviations were correct in original version. They were in left upper case and probably disappeared upon conversion of WORD format into PDF format that you have received. The abbreviations are still as they were.

AMENDMENT:

We have removed atom numbering from Scheme 1 (which now shows only the synthetic path and formula view). We have transferred the PAMAM main chain atom numbering at Figure 1 for reader’s convenience. Other Figures with 1-H and 13-C spectra will need the atom numbering as well, but we cannot repeat the atom numbering everywhere.

Anyway, thank you very much for suggestion. Eventually it will make paper more readable.

  1. How can you determine the amine residue of G2 or G3 after modification? Please double check with quantitative determination of amine.

ANSWER AND EXPLANATION:

The starting PAMAM G2 and G3 dendrimers have perfect structure with 16 and 32 amine groups, respectively. We synthesize them once a year, which takes ca 6 weeks to properly purify them by removal of ethylenediamine used in excess. We keep the compound as solution in methanol and check their purity just before modification. Due to Reviewer 2 remark we added short description of PAMAM dendrimers synthesis (Section 2.1.). Anyway, the compounds are commercially available.

The complete addition of glycidol into G2 and G3 results in every surface amine group substituted with 2,3-dihydroxypropyl residues. The integration of multiplet of AB-type from terminal methylene group protons of 2,3-dihydroxypropyl residues is compared with well separated methylene group protons next to carbonyl group of PAMAM dendrimer, which is used as internal intensity reference. Its intensity corresponds to [56H] for G2 and [120H] for G3. Thus the integral intensity of 2,3-dihydroxypropyl proton resonances gives the straightforward quantitative answer to the question how many 2,3-dihydroxypropyl groups are attached to G2 or G3. For fully substituted G2 and G3 (see Figure 1B), the integration of terminal gly-derived methylene group (g) gives number 64.55, while for broad multiplet resonance for proton f the integral is 30.78. These are the output numbers of integration without any manipulation with baseline, just row data and reproducible range of integration for every spectrum. According to my 40 years NMR experience these numbers are close to 32 and 16, which are theoretically the right numbers for G232gl.

When deficient number of glycidol residues is used on purpose to obtain not totally glycidol-covered G2 or G3, the analogous integration procedure is reproducibly applied (Figure 3). For G2 the comparison of integral intensity of reference signal at 2.42 ppm [56H] with g protons of 2,3-dihydroxypropyl residues gives the number of 9 (18.20 integral intensity), 14 (29.16 integral intensity), and 22 (44.22 integral intensity). It should be emphasized that the stoichiometry is an AVERAGE for every species. We realize that G222gl is actually the mixture of species with variable amount of gl residues, with maximum at 22. The same is for other species isolated. The GPC which we did for deficiently substituted species clearly gives broader peaks than GPC for fully substituted G232gl and G364gl. The latter two, together with G4128gl and G5256gl (where gl was used as racemate) served us before as reference compounds for GPC studies, size distribution by DLS, and zeta potential (that results were published in [21] in original version of paper, now cited as [XX]). These references enabled us to define the stoichiometry of mixed conjugates, composed of covalently attached G3 and G0 with substituents [21].

The same 1-H NMR signal intensity procedure was then applied for fluorescein- and rhodamine-substituted G2 and G3 after conversion with S- and R-glycidol. Here we could determine the number of glycidol equivalents attached into core dendrimer. Unfortunately, you are right about the number of FREE amine groups. They cannot be determined exactly because some arms are substituted  with two glycidol equivalent and some are substituted with one glycidol equivalent. 1-H NMR spectra of both kind of glycidol residues are almost the same, although 13-C spectra of double-substituted arms and single-substituted arms are clearly distinguishable. This spectral characteristic is described in Section 3.1.4 without trying to speculate on number of “free” amine groups. In fact we searched for total spectral assignment for G232Racgl and we failed to make it due to 13-C NMR resonances were not well-resolved at 300 MHz instrument.

  1. In case of comparison between PAMAM-pyrrolidone and other ones, could you summarize the results with citations, not just write “… highly satisfactory (see ref [30]”?

AMENDMENT:

Appropriate text was added (lines 463-467): For example, considered to be low-toxic PAMAM G4 – pyrrolidone dendrimer was toxic already after 2 or 6 hours incubation with 200 µM concentrations against mouse neuroblastoma (N2a) cells (viability lowering for about 40%) [31]. A 25% decrease in viability of mHippoE-18 embryonic mouse hippocampal cells was also seen after 200 µM concentration treatment [32] . 

  1. How about the molecular weight of G2, G3 and modified G2, modified G3? You should operate GPC for your polymer to strongly confirm your synthesized polymer beside NMR.

Yes, we did the GPC, LDS, and zeta measurements as described in [23], for details please see:

Zaręba M, Sareło P, Kopaczyńska M, Białońska A, Uram Ł, Walczak M, Aebisher D, Wołowiec S. Mixed-Generation PAMAM G3-G0 Megamer as a Drug Delivery System for Nimesulide: Antitumor Activity of the Conjugates Against Human Squamous Carcinoma and Glioblastoma Cells. Int. J. Mol. Sci. 20 (2019) 4998. doi:10.3390/ijms20204998.

Other details on GPC, DLS and zeta potential are given in Answer to point 10.

Reviewer 2 Report

Paper titled (Chiral recognition of homochiral PAMAM dendrimers substituted with R- and S-glycidol by keratinocyte (HaCaT) and squamous carcinoma (SCC 15) cells in vitro) by Malinga-Drozd et al., demonstrated the recognition of substituted PAMAM densrimers by HaCaT and SCC15. It is a novel study but needs some revisions.

I have the following comments:

1-Titles: better to say Polyamidoamine  not   PAMAM. 
Also define it at the first appearance in abstract.

please clarify at the first part of abstract and iNtro how authors proved chiral recognition

2- Introduction is too long; can be shortened to be more concrete and focus on the topic

3-PAMAM dendrimers synthesis in 2.1. need to be explained

4- Kindly justify, why the toxicity study was done for 24 h only

5-Fig 5: any IC50 values were calculated?

6-Authors have to mention the type of the presented data in each illustration starting from Fig 5 (for example mean +SD or SE...etc)
7- Kindly write as Kruskal Waliis ANOVA not test

8-Authors have to justify why they used Kruskal-Wallis ANOVA? did you check the goodness of fitness by a special test such as KS test

9- Kindly describe how images were processed 

10- conclusion is too long as well

Author Response

REVIEWER 2

Paper titled (Chiral recognition of homochiral PAMAM dendrimers substituted with R- and S-glycidol by keratinocyte (HaCaT) and squamous carcinoma (SCC 15) cells in vitro) by Malinga-Drozd et al., demonstrated the recognition of substituted PAMAM densrimers by HaCaT and SCC15. It is a novel study but needs some revisions.

I have the following comments:

1-Titles: better to say Polyamidoamine  not   PAMAM. 
Also define it at the first appearance in abstract.

please clarify at the first part of abstract and iNtro how authors proved chiral recognition

ANSWER:

Thank for your suggestions:

AMENDMENTS (in blue):

Chiral recognition of homochiral poly(amidoamine) dendrimers substituted with R- and S-glycidol by keratinocyte (HaCaT) and squamous carcinoma (SCC 15) cells in vitro

Abstract (added text in blue, lines 13-17 in the revised version):

The generation 2 and 3 poly(amidoamine) dendrimers (PAMAM G2 and G3) were converted into N-(2,3-dihydroxy)propyl derivatives by addition of enantiomerically pure S- and R-glycidol. The homochiral dendrimers bind to cell membrane (HaCaT and SCC-15) with R/S glycidol enantioselectivity ratio 1.5:1 as was quantitatively determined by fluorescence microscopy and visualized by confocal microscopy.

2- Introduction is too long; can be shortened to be more concrete and focus on the topic

ANSWER:

The remark is important, however it is in contrary to Reviewer 1 suggestion, who obliged us to add new citations on recent developments in dendrimer chemistry for medical purposes. So we have introduced two new citations.

In the introduction we intended to show the importance of macromolecular chiral space imposed by chiral dendrimers. We have addressed this paper not only to polymer chemists (the journal title is Polymers), but also to those who use the chiral space for asymmetric catalysis (the lines 55-86 in the revised version). We consider this important for Polymers journal which IF is growing due to general attention. On the other hand other cited papers are related to chiral dendrimers and biological recognition. Our paper is in fact the first that shows the enentioselectivity of homochiral dendrimers in cell binding and internalization . We consider this an important conclusion, but please leave the Introduction as such, just for readers not involved in biological applications.

3-PAMAM dendrimers synthesis in 2.1. need to be explained

ANSWER:

Although PAMAM dendrimers are commercially available we cannot afford to use them at the multigram scale. Therefore once a year we synthesize them on 30 g scale of G3 leaving intermediates in large amount as well for other studies. The synthetic procedure was elaborated by Tomalia in early 80s. We shortly introduced the one sentence on PAMAM syntheses according to the suggestion.

AMENDMENT (in blue, lines 121-125):

PAMAM dendrimers were synthesized by alternate addition of methyl acrylate into amine groups, starting from ethylenediamine core, followed by condensation with ethylenediamine, purified intermediates at every step according to protocol of Tomalia [1] and stored as 20 weight % solution in methanol.

4- Kindly justify, why the toxicity study was done for 24 h only

ANSWER:

In our previous studies of dendrimers, we examined the toxicity of the dendrimers at 24, 48 and 72 hours. Typically, the toxicity at 24 hours was clearly evident, and extending the incubation time did not increase it drastically. Moreover, in the literature concerning dendrimers, the 24-hour incubation period is the most frequently considered, and therefore we can relate the results of our research to the results of other authors. For these reasons, we decided to test the toxicity after a 24-hour incubation.

5-Fig 5: any IC50 values were calculated?

ANSWER:

We did not calculate the IC50, because the stock solutions of studied dendrimers in water was equal  3 mM. Therefore, the highest possible working solution was 300 micromolar to avoid the influence of osmotic pressure changes. At this concentration, there were no signs of toxicity against any of the tested cell lines, therefore it was not possible to determine the IC50 coefficient. It is also worth mentioning that the lack of toxicity at the level of 300 micromolar concentration makes the compound non-toxic and highly biocompatible. Therapeutic blood concentrations of carriers and drugs are usually significantly below 300 moles.

6-Authors have to mention the type of the presented data in each illustration starting from Fig 5 (for example mean +SD or SE...etc)

ANSWER:

Thank you for your valuable remark. We have used a non-parametric statistical tests, therefore we have presented results as medians and the whiskers on figures are lower (25%) and upper (75%) quartile ranges. An appropriate information was added in Figure 5 and 6 descriptions (lines 450 and 510-511).

7- Kindly write as Kruskal Waliis ANOVA not test

and

8-Authors have to justify why they used Kruskal-Wallis ANOVA? did you check the goodness of fitness by a special test such as KS test

ANSWER and AMENDMENT (to point 7 and 8):

Thank you for your valuable remark. We chose the non-parametric test due to the lack of a normal distribution of data in the studied groups (analysis with Shapiro-Wilk test). The proposed Kolmogorov Smirnov test is just as appropriate as the Shapiro-Wilk test. Information about Shapiro-Wilk test was added in Materials and Method – Statistical analysis part (lines 230-231).

9- Kindly describe how images were processed 

ANSWER and AMENDMENT:

Raw confocal microscopy images from Olympus FV10i were collected and saved in .oif format. Then the processing was carried out in ImageJ: the channels were assigned specific colors (edit LUT function) and channels were merged (Merge Channels function). Then gallery was made. Additional information has been added in the materials and methods section (lines 219-220).

10- conclusion is too long as well

ANSWER:
We have removed the sentences which were already signaled in Results and Discussion part.

AMENDMENT (removed part):

Obtained results demonstrated that G3gl and G2gl furnished with R-glycidol may be an excellent candidate as drug delivery system into squamous cell carcinoma, depending on the destination of the drug transferred. For supramolecular megameric system, the PAMAM G5 fully substituted with R-glycidol will be used as core dendrimer, to which PAMAM G0 bearing combination of drugs will be attached covalently as we presented before for G3gl-G0 megamers containing nimesulide [23].
